# Effect of Heat Treatment Condition on Microstructural and Mechanical Anisotropies of Selective Laser Melted Maraging 18Ni-300 Steel

**Dohyung Kim [1,2], Taehwan Kim [1,2], Kyeongsik Ha [1,3], Jeong-Jung Oak [4], Jong Bae Jeon [1], Yongho Park [2,\*] and Wookjin Lee [1,\*]**

[1] Dongnam Regional Division, Korea Institute of Industrial Technology (KITECH), Yangsan 50623, Korea; dhyungkim@kitech.re.kr (D.K.); hlight@kitech.re.kr (T.K.); hamazzang123@kitech.re.kr (K.H.); jbjeon@kitech.re.kr (J.B.J.)

[2] Department of Materials Science and Engineering, Pusan National University, Busan 46241, Korea

[3] Department of Mechanical Engineering, Pusan National University, Busan 46241, Korea

[4] Material Analysis Laboratory, Dea-Il Corporation, Ulsan 44914, Korea; ojj69@dicorp.co.kr

\* Correspondence: yhpark@pusan.ac.kr (Y.P.); wkjinlee@kitech.re.kr (W.L.); Tel.: +82-55-367-9409 (W.L.)

**Abstract:** 18Ni-300 maraging steel produced by the selective laser melting (SLM) process has a unique microstructure that is different from that of the same alloy processed by conventional methods. In this paper, maraging steels were fabricated by the selective laser melting process and their microstructures and mechanical properties were investigated in terms of post heat treatment conditions. Moreover, the effect of different heat treatments on the mechanical anisotropy was studied in detail. The micro Vickers hardness in the as-built state was around 340 Hv and could be increased to approximately 600 Hv by aging heat treatments. It was found that the solution heat treatment was not necessary to obtain a fully hardened state. From tensile tests of the maraging steels heat treated with different conditions, it was found that the highest strength was achieved by aging and solution treatment (ST) temperatures lower than the commonly used temperatures. In the direction parallel to the laser scanning, the highest ultimate tensile strength was obtained when 450 °C aging was done without solution heat treatment. In the other two directions tested, i.e., directions normal to the building and 45 degrees to the laser scanning direction, the highest tensile strength was obtained when aging was done at 450 °C after 750 °C solution treatment.

**Keywords:** maraging steel; selective laser melting; mechanical properties; microstructure; heat treatment

## 1. Introduction

Selective laser melting (SLM) is one of the layer additive manufacturing processes for metallic parts that fabricates complex parts by selectively consecutive layers of powder particle using a laser beam. A high-power laser is used as a heat source to melt and fuse specific regions of the powder that is spread out on a powder bed. Due to good material efficiency and design flexibility, the SLM process can be used to fabricate customized parts, tooling inserts with conformal cooling channels, and functional components with high geometrical complexity in relatively small quantities [1,2].

Maraging steels are a special class of low-carbon and high-strength steels that can be hardened by a martensitic phase transformation occurring after an aging heat treatment [3]. They have been used in a wide range of applications including casting molds, aircraft components, and tool die materials where ultra-high strength and high hardness are required. Fine $Ni_3(Al, Ti, Mo)$ and $Fe_2Mo$ intermetallic particles are precipitated during aging treatments at 500 to 550 °C, which result in high hardness and strength of these alloys [4,5]. Because strengthening occurs mainly by intermetallic precipitates and not

by high carbon content, they generally show much higher toughness than conventional high-carbon martensite alloys. Additionally, some advantageous properties make them attractive for the fabrication of machinery components by SLM. For instance, the extremely low carbon content of maraging steels facilitates their application in the SLM process because no special care is needed to avoid carbide or carbon segregation-related problems.

Preliminary studies regarding the maraging steels produced by SLM showed that they have a unique microstructure and unique mechanical properties in comparison with those of the conventionally processed ones. For instance, Takata et al. [6] investigated the microstructure of maraging steel produced by SLM and found that the alloy had a martensite phase in the as-built state due to an extremely fast cooling rate of up to $10^6$ °C/s. It has been also shown that the maraging steel fabricated by SLM has finer structures compared to those of the conventional manufacturing methods due to a sufficiently fast cooling rate, resulting in higher strength [7]. Another significant difference in microstructures of maraging steels produced by SLM and conventional routes is that nuclei of the precipitates can be easily formed during the SLM process. The alloy produced by SLM undergoes reheating repeatedly while neighboring tracks and subsequent layers are deposited [7–9]. This repetitive heating can act as an intrinsic heat treatment during the process, which facilitates the nucleation of intermetallic precipitates without post heat treatment. Therefore, the precipitation can occur easily without solution heat treatments. In this regard, some researchers have already anticipated to skip the solution heat treatment for the maraging steel produced by SLM. For instance, Kempen et al. [10] have recently revealed that excellent tensile properties can be achieved by only performing an aging heat treatment without solutioning for maraging steels fabricated by SLM. In their study, both as-built and aged maraging steel samples consisted of a martensite phase greater than 96%. After aging treatment, the hardness and tensile strength were dramatically increased by precipitations of intermetallic particles. Yin et al. [11] and Tan et al. [9] investigated the effects of aging time and temperature on the mechanical properties of maraging steels fabricated by SLM, with and without the solution heat treatment. They showed that the properties of maraging steels depend largely on the aging temperature rather than the aging time. It was also found that the fracture elongation of maraging steel that underwent aging directly without solutioning was much smaller than that experienced with the conventional heat treatment. Therefore, more attempts are required to obtain the optimal heat treatment condition for maraging steels fabricated by SLM.

Another important aspect of the materials produced by SLM is their strongly anisotropic mechanical properties. Due to the unique microstructures developed during the SLM process, which are composed of a number of directionally solidified laser beads, the material often has completely different mechanical properties in different directions. It has been reported that the SLM-processed Inconel [12], AISI 316L stainless steel [13,14], and titanium alloys [15] have strongly anisotropic mechanical properties. The mechanical anisotropy of the maraging steel produced by SLM has also been recently studied by Suryawanshi et al. [16] for AISI 18Ni-300 maraging steel. However, they only studied the mechanical anisotropy of the alloy in the as-built state. So far, there has been a lack of studies on anisotropy of the maraging steel produced by SLM in the heat-treated condition.

In this paper, AISI 18Ni-300 maraging steel was fabricated by SLM and the effect of heat treatments on its microstructure and mechanical anisotropy was studied. Different heat treatment conditions were used, i.e., aging with or without solutioning at different temperatures, and the results were compared to those with the standard heat treatment conditions that have been frequently used in the literature [9,17].

## 2. Materials and Methods

Gas-atomized maraging 18Ni-300 steel powder (OPM Maraging, OPM Laboratory Co., Ltd., Kyoto, Japan) was used as the starting material. The morphology of the powder was investigated by scanning electron microscope (SEM, JSM-7200F, Jeol Inc., Tokyo, Japan). As shown in Figure 1a, the particles of the powder were mostly spherical with few irregular shapes. Figure 1b shows the particle

size distribution, measured using a laser diffraction particle size analyzer (LS 13-320, Beckman Coulter, Miami, FL, USA). The powder had a bimodal particle size distribution with modal diameters centered at 7 and 40 µm, to obtain a good packing density.

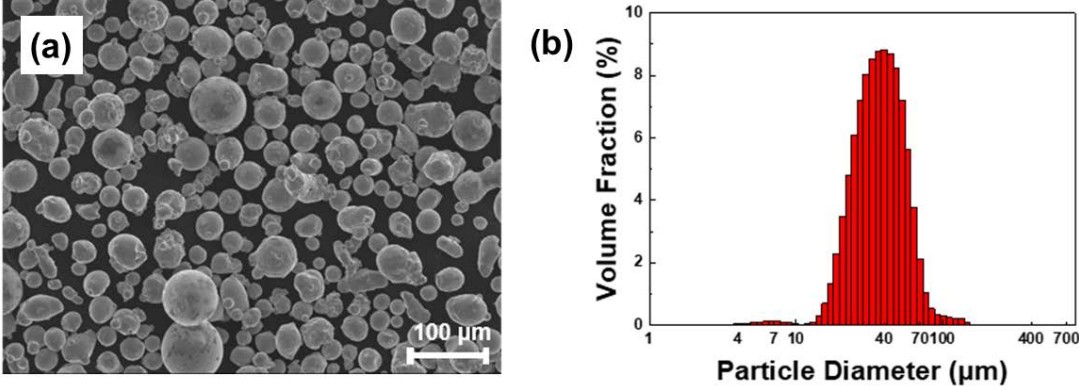

**Figure 1.** (**a**) Scanning electron microscope (SEM) image of maraging steel powder used for selective laser melting and (**b**) its particle size distribution.

The samples were manufactured using an SLM type metal 3D printer (OPM250L, Sodick Co., Ltd., Kyoto, Japan), equipped with single-mode ytterbium fiber laser with a wavelength of 1,070 nm and a maximum laser output of 500 W (YLR-500-WC, IPG, Laser GmbH, Burbach, Germany). The SLM of the specimen was done under a laser power of 420 W, a scanning speed of 1,000 mm/s, and a hatch spacing of 0.1 mm with a lamination thickness of 0.04 mm. A so-called 90° rotate scanning strategy was used, i.e., the laser scanning lines are 90° tilted between each layer. Figure 2a shows schematically the scanning strategy used for the SLM. The selective laser-melted samples had a cubic shape with dimensions of $50 \times 50 \times 50$ mm$^3$, and then were cut into small-sized tensile specimens with a plate shape according to the KS B0801 standard [18]. The gauge length, width, radius of fillet, and length of the reduced section were 9.8, 2, 4, and 13.2 mm, respectively. To study the anisotropy of the tensile property of the samples, three types of specimens were cut at different directions relative to the SLM building axis, as shown in Figure 2b. The specimens having a tensile direction parallel to the building direction were designated as BD. The two other types of specimens had tensile directions normal and 45° to the laser scanning directions. These specimens were designated as SD and 45SD, respectively.

Post heat treatments of the specimens were performed in a box-type laboratory furnace. The specimens were wrapped in a protective heat-treatment foil to prevent oxidation of the sample surface. Some specimens underwent solution treatment (ST) for 2 h at two different temperatures of 750 and 850 °C and then aged at various temperatures of 400, 450, and 500 °C for 6 h. Other specimens were directly aged without ST at 450 °C for 6 h. Tensile tests were performed for the as-built and heat-treated specimens in a universal testing machine (RB 301 UNITECH-T, R&B Inc., Daejeon, Republic of Korea), with a constant loading rate of 0.1 mm/min. Strain gauges were attached to the specimens for elongation measurements.

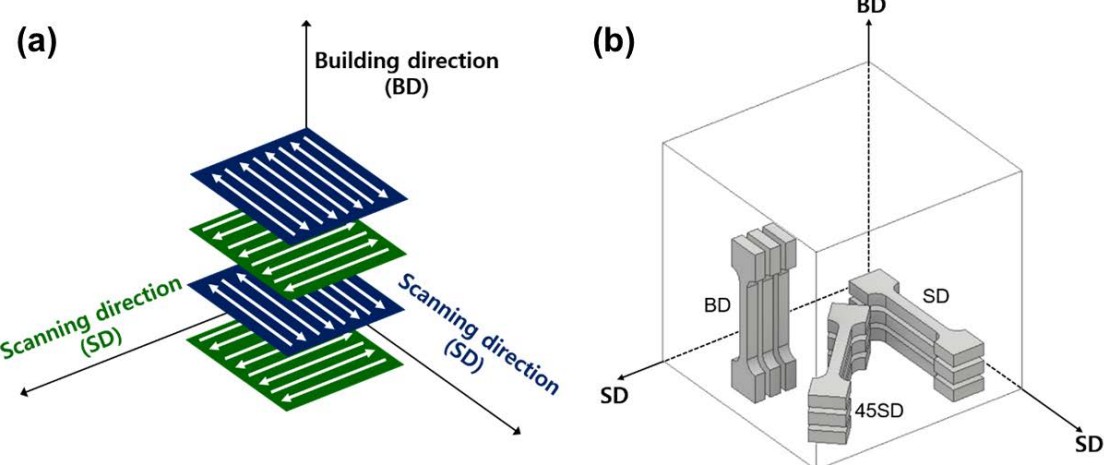

**Figure 2.** Schematics of (**a**) laser scanning strategy and (**b**) sample building direction (BD) for selective laser melting of maraging steel samples.

Some test pieces were cut from the cubic block fabricated by SLM and underwent post heat treatments with the same conditions as used in the tensile specimens. The test pieces were ground and polished in order to observe the microstructure. Using the test pieces, phase compositions were determined using an X-ray diffractometer (XRD, EMPYREAN, PANalytical BV, Almelo, Netherlands) with Cu $K\alpha$ radiation over a two-theta range from 20° to 80°. Vickers microhardness (Hv) was measured as the average of 5 separate tests at the surface of each specimen under a load of 0.1 kgf (HM-220A, Mitutoyo Corp., Kawasaki, Japan). The microstructures of the test pieces were investigated by optical microscope (OM, Eclipse E200, Nikon, Tokyo, Japan), SEM, and electron backscatter diffraction (EBSD, Aztec HKL, Oxford Inc., Abingdon, UK).

## 3. Results and Discussions

Figure 3 shows the hardness of the test pieces of the maraging steel with different building directions and heat treatment conditions. In the as-built conditions, the hardness of the test pieces remained practically the same regardless of the building direction, at around 340 Hv. This indicates that the effect of the mechanical anisotropy is negligible to the microhardness of the maraging steel fabricated by SLM.

In general, the hardness of the test pieces increased significantly with the heat treatments, as shown in Figure 3. This behavior can be easily understood from the strengthening mechanism, i.e., by a uniform distribution of fine nickel-rich intermetallic precipitates during the aging of a ductile, low-carbon martensite structure, as previously shown by many other studies [4,8,9,11]. When a relatively low aging temperature of 400 °C was used, the hardness of the test pieces was raised to 475–545 Hv after 6 h aging, depending on the ST conditions. There was a pronounced difference in hardness among the test pieces that underwent post heat treatments with different ST conditions before the aging. The hardness of the test piece with ST at 850 °C for 2 h before the aging showed the lowest hardness of ~475 Hv, whereas the highest hardness was observed in the test piece treated by 750 °C ST before the aging, at ~545 Hv. When the aging temperature was raised to 450 °C, the hardness of the test pieces increased further, close to 600 Hv. Under this condition, only a slight difference existed in the hardness among the test pieces with different ST conditions, i.e., the sample treated by 850 °C ST showed lower hardness than the other samples by ~20 Hv. The effect of ST condition became negligible with the increased aging temperature of 500 °C, where the hardness of all the test pieces was around 600 Hv.

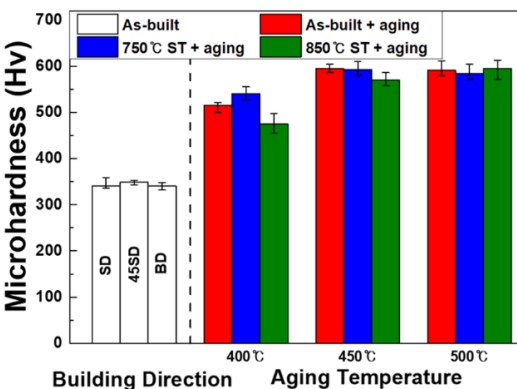

**Figure 3.** Effect of building direction and heat treatment condition on hardness of maraging steel fabricated by selective laser melting (SLM). Note that the hardness of the aged test pieces was evaluated on the scanning direction (SD).

The results of hardness measurements showed that, almost regardless of the ST condition, a hardness value close to 600 Hv can be obtained by aging at 450 °C for 6 h. The 600 Hv value can be considered as the hardness of the fully hardened maraging steel fabricated by SLM, as no additional hardening effect was observed after increasing the aging temperature to 500 °C, as can be seen in Figure 3. This indicates that an almost fully hardened state is achievable with an aging temperature of 450 °C for maraging steel fabricated by SLM. It is worth noting that the aging temperature of 450 °C is lower by 30–40 °C in comparison with the aging temperature used for the maraging steel fabricated by the conventional manufacturing process and SLM [6,8].

The OM images in Figure 4 present the microstructural changes during the different heat treatments of the maraging steel samples fabricated by SLM. Figure 4a shows the macroscopic microstructure in the as-built condition. The melt pool boundaries caused by the 90° rotate scanning strategy can be clearly seen in the microstructure where laser scan tracks of every second layer run parallel. When only the aging heat treatment was done at 450 °C without ST, the melt pool boundaries remain clearly visible, as can be seen in Figure 4b. The melt pool boundaries became less noticeable when 750 °C ST was done before the aging, as shown in Figure 4c. The boundaries were no longer visible after ST at 850 °C (Figure 4d).

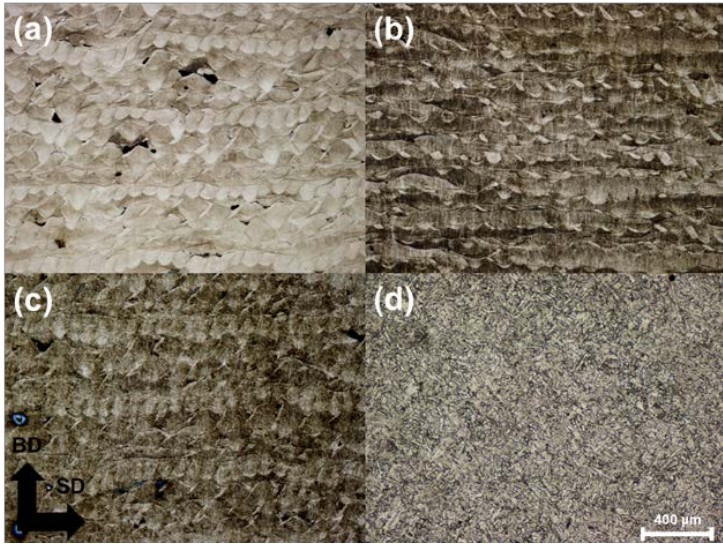

**Figure 4.** Optical microscope (OM) images of maraging steel samples (×50): (**a**) as-built condition; (**b**) after 6 h aging at 450 °C without ST; (**c**) after 750 °C ST for 2 h and aging at 450 °C for 6 h; and (**d**) after 850 °C ST for 2 h and aging at 500 °C for 6 h.

Figure 5 shows higher magnification OM images of the same microstructures shown in Figure 4. As can be seen in Figure 5a, the height of the bead was approximately 40 μm, which is equal to the lamination thickness used in the SLM process. The width of the bead was about 150 μm. In the microstructures of the as-built and the heat-treated samples, coarse precipitates are visible in the microstructures, some of which are indicated by white arrows in Figure 5. In the as-built condition, few precipitations occurred along the melt pool boundaries, as shown in Figure 5a. In the sample aged at 450 °C without ST, some spherical precipitates are also observed both inside the bead and near the melt pool boundaries (Figure 5b). When 750 °C ST was done before the aging, further coarsened spherical precipitates were observed in comparison with the sample without the ST, as shown in Figure 5c. Finally, 850 °C ST and 500 °C aging produced lath-type microstructure by fine precipitates with a few spherical precipitates, as shown in Figure 5d.

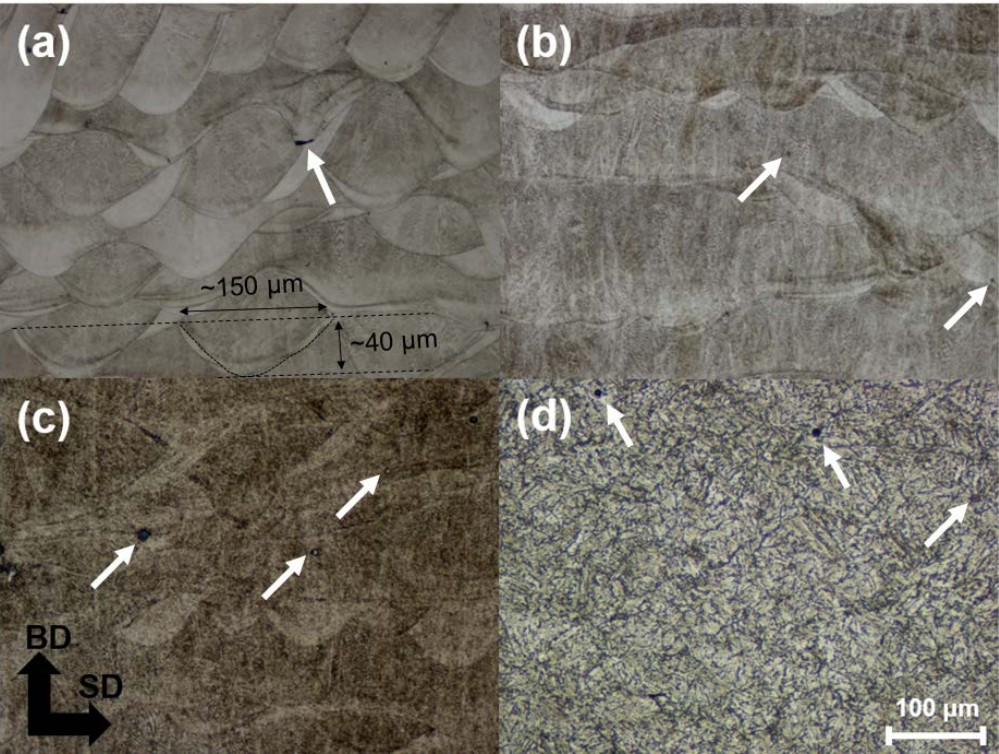

**Figure 5.** OM images of maraging steel samples (×200): (**a**) as-built condition; (**b**) after 6 h aging at 450 °C without ST; (**c**) after 750 °C ST for 2 h and aging at 450 °C for 6 h; and (**d**) after 850 °C ST for 2 h and aging at 500 °C for 6 h.

Figure 6 shows high-magnification SEM images (×2500) of the maraging steel samples. Fine cell structures are clearly visible in the microstructures of the as-built and the sample with only aging at 450 °C without ST. It was found that the cell size became slightly larger with the aging heat treatment at 450 °C, as shown in Figure 6a,b. The samples that underwent the ST before the aging heat treatment did not show a clear cell structure. Very fine intermetallic precipitates are clearly observed in the sample aged at 450 °C with 750 °C ST, as shown in Figure 6c, whereas comparably large precipitates are shown in the sample aged at 500 °C with 850 °C ST (Figure 6d).

In order to identify the chemical composition of the coarse precipitates observed in Figure 5, energy dispersive X-ray mapping was performed. Figure 7a–e shows the chemical composition maps near the precipitate formed along the melt pool boundary in the as-built condition. It clearly shows that the precipitate contained no other metallic elements nor carbon except a high concentration of titanium. A similar observation was made by Kang et al. [19], who found a Ti-rich region near melt pool boundaries in SLM-processed maraging steel. One possible explanation for the formation of

this precipitate is the segregation of titanium at the melt pool boundary during rapid solidification in the SLM process. The spherical precipitate formed during the heat treatment is also identified as the titanium-rich phase, as shown in Figure 7f–j. Although the microstructural origin of this type of precipitate is still not fully understood, it is known that the formation of this precipitate is harmful to the mechanical properties of the SLM-processed maraging steel as it can act as the crack initiation site [20]. In Figure 7f, some spherical holes are also observed which are thought to be due to the removal of the spherical precipitates during the sample preparation. The lath-shaped fine precipitates are also distinguished in Figure 7f, which are presumably nickel intermetallic compounds such as $Ni_3Al$, $Ni_3Ti$, $Ni_3Fe$, or $Ni_3Mo$ [8,9]. However, no significant difference in the chemical composition maps could be found near the lath-shaped precipitates, probably because of their small size.

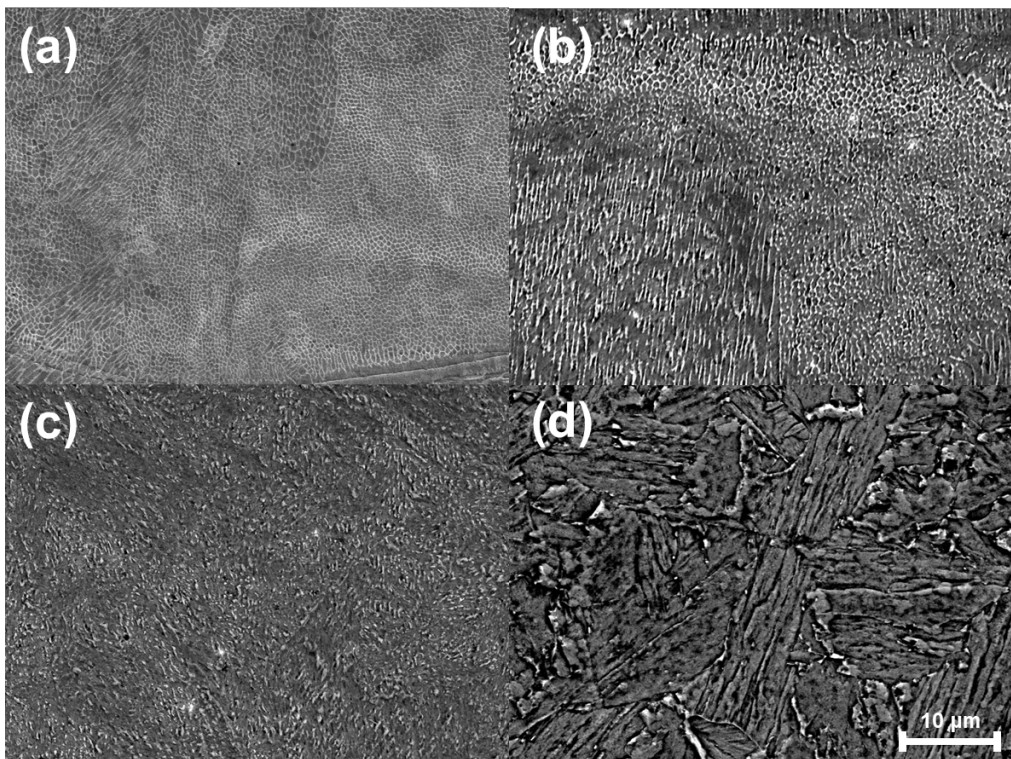

**Figure 6.** SEM images of maraging steel samples (×2500): (**a**) as-built condition; (**b**) after 6 h aging at 450 °C without ST; (**c**) after 750 °C ST for 2 h and aging at 450 °C for 6 h; and (**d**) after 850 °C ST for 2 h and aging at 500 °C for 6 h.

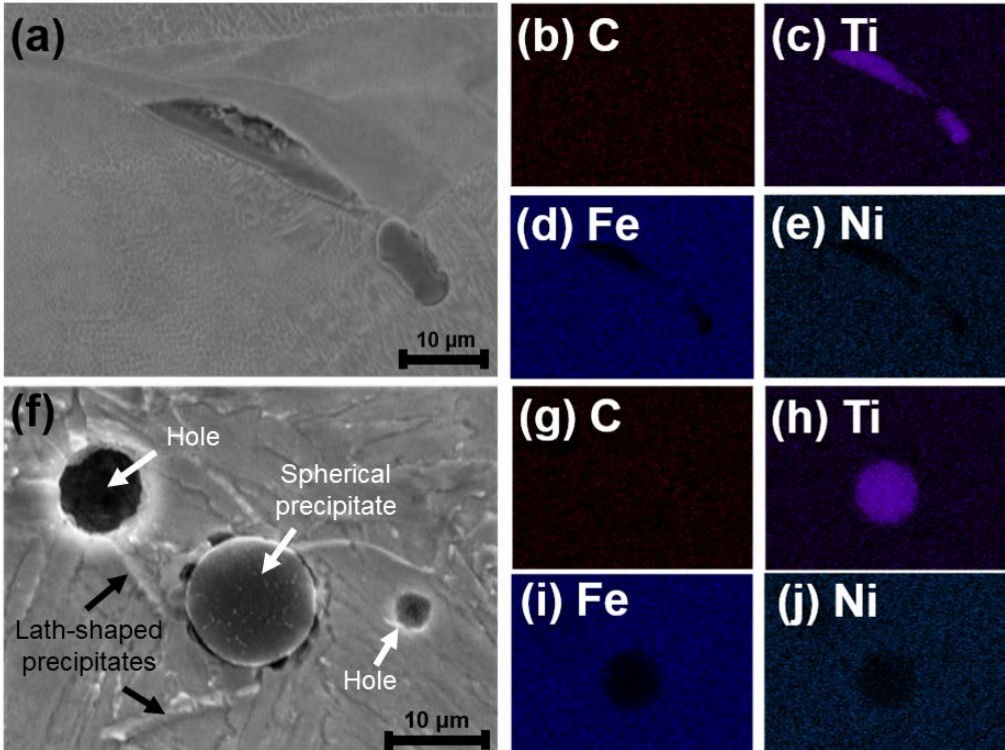

**Figure 7.** (**a**) SEM image and (**b–e**) energy dispersive X-ray mappings of coarse precipitate formed along melt pool boundary in as-built condition; (**f**) SEM image and (**g–j**) energy dispersive X-ray mappings of spherical precipitate inside bead after 850 °C ST for 2 h and aging at 500 °C for 6 h.

Figure 8 shows XRD patterns for the as-built and differently heat-treated maraging steel samples fabricated by SLM. The major diffraction peak shown in the patterns correspond to the $\alpha'$-martensite (body-centered cubic) phase for all the samples including the as-built sample, indicating that the maraging steel produced by SLM has an almost full martensite microstructure in the as-built condition due to the rapid cooling. There is a small $\gamma$-austenite (face-centered cubic) peak in the as-built, 450 °C aged without ST, and 450 °C aged after 750 °C ST samples, reflecting that the amount of residual austenite is small in the microstructure. which disappears when the sample was aged at 500 °C after 850 °C ST. There are other small peaks corresponding to the nickel intermetallic compounds such as $Ni_3Al$ ((001)-24.94°, (101)-38.52°), $Ni_3Ti$ ((101)-22.84°), $Ni_3Fe$ ((001)-25.08°, (101)-38.75°), or $Ni_3Mo$ ((101)-26.71°, (200)-35.48°), at the positions marked by star signs in Figure 8. The exact type of the precipitate was difficult to be distinguished from the XRD patterns since the main peaks are close to each other. However, all the small peaks in the range between 20° and 40° are expected to represent nickel intermetallic compounds. These peaks were most clearly seen in the sample aged at 450 °C without ST, but the same peaks were also noticeable in all the other samples. No additional peak occurred that could be indexed to the Ti-rich precipitates, probably due to their small volume fraction.

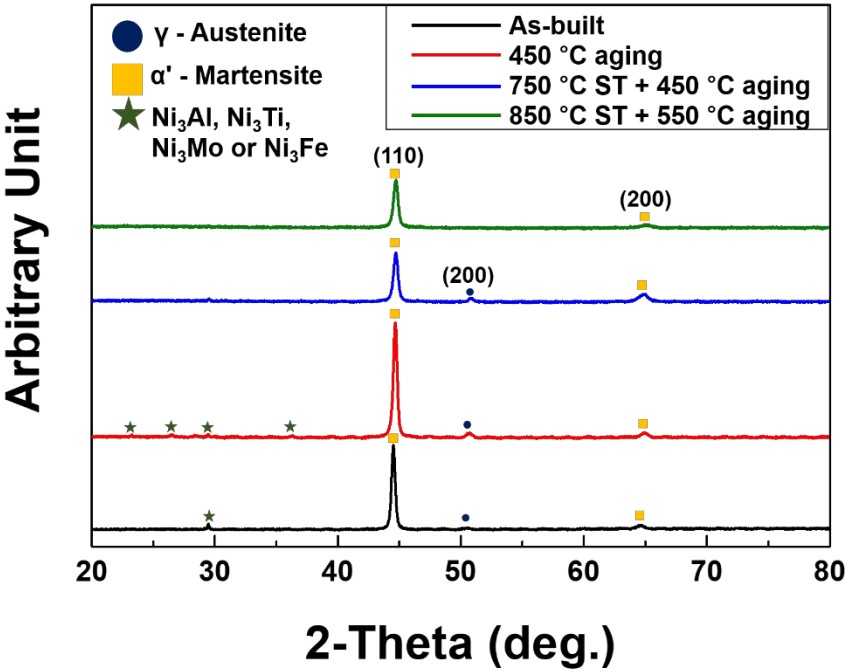

**Figure 8.** X-ray diffraction (XRD) patterns of as-built and heat-treated maraging steel samples fabricated by SLM.

The crystallographic texture analysis was conducted using the EBSD patterns and pole figures for the maraging steels produced by SLM with different heat treatment conditions. For the $\alpha'$ phase, EBSD BD-inverse pole figure maps and (100), (101), and (111) pole figures are presented in Figure 9. The inverse pole figure maps shown in Figure 9a-d represent the orientation of the grains along the BD and the center of all pole figures given in Figure 9e–h is parallel to the BD of the maraging steel samples produced by SLM. The texture in the as-built condition presented in Figure 9a,e shows that a clear <100> as well as <110> alignment occurred in the building direction during the SLM process. This reveals that the $\gamma$ phase texture was formed during the solidification and then crystallographic transformations from $\gamma$ to $\alpha'$ phase happened with variant selections. When only 450 °C aging heat treatment was done without ST, the texture was retained almost completely, as shown in Figure 9b,f. The texture was significantly weakened when ST was done before aging treatment, as is clearly shown in Figure 9g,h.

When ST was done at 750 °C, there was no pronounced grain growth or grain rearrangement, as observed by comparing the microstructures shown in Figure 9a–c. However, when the sample was aged at 500 °C after 850 °C ST, the former complex microstructure that occurred by the SLM process disappeared and coarser martensite grains were produced, as shown in Figure 9d. This may indicate the recrystallization of the $\gamma$ phase during the ST at 850 °C.

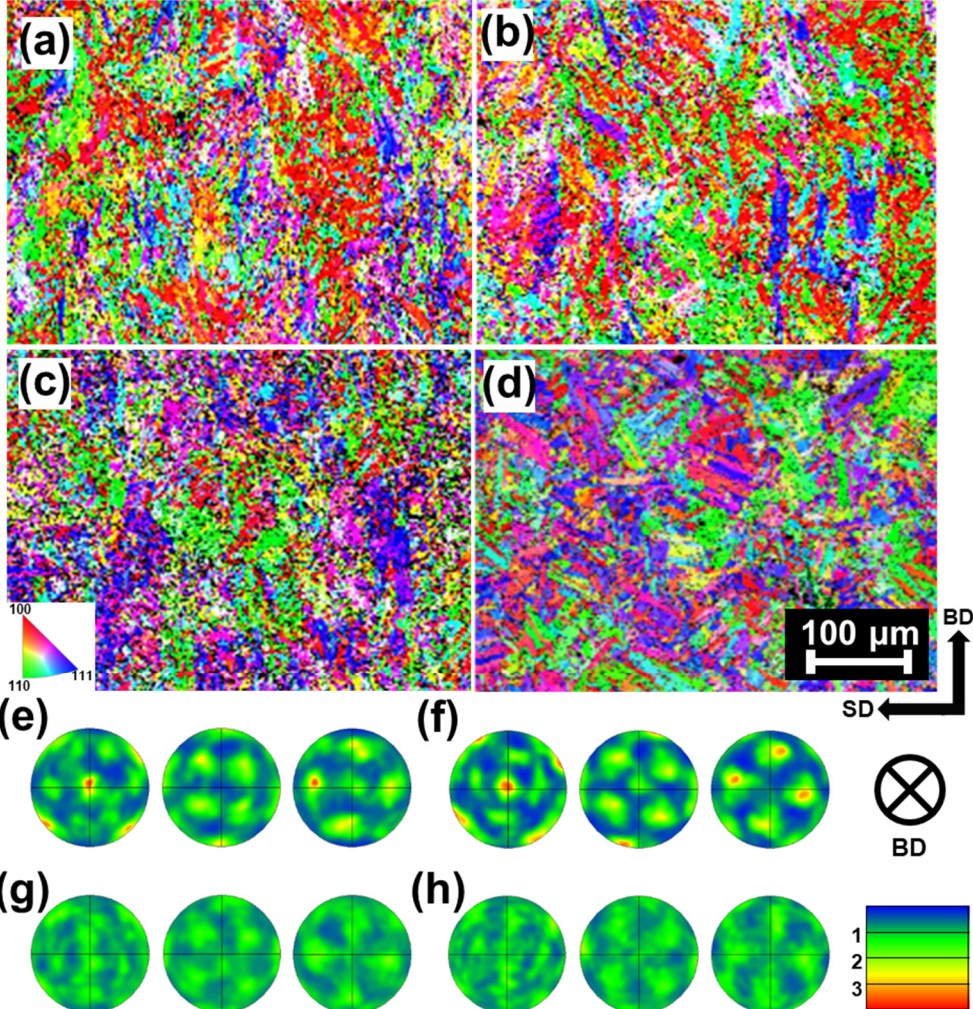

**Figure 9.** (**a–d**) Electron backscatter diffraction (EBSD) BD inverse pole figure maps and (**e,f**) pole figures. The center of the pole figures coincides with the building direction: (**a,e**) as-built; (**b,f**) 450 °C aged without ST; (**c,g**) 450 °C aged after 750 °C ST; and (**d,h**) 500 °C aged after 850 °C ST.

Figure 10 shows the tensile stress–strain curves for the maraging steels produced by SLM with different heat treatments, along with different directions. For all tensile directions tested, the samples in the as-built condition were characterized by relatively high ductility and low strength, with elongations at break of 8.5–9.5% and yield strength of around 1,000 MPa. Heat-treated samples generally exhibited very high strength of up to 2.1 GPa and low elongation less than 3%, indicating that improvements in the alloy's strength were achieved by fine nickel-rich intermetallic precipitates upon aging heat treatments [8,9]. As shown in the figure, it is evident that the heat treatment affects differently the stress–strain behavior of the different sample directions. Along the SD, the yield strength of the maraging steel samples was the highest when only the aging heat treatment was performed at 450 °C without ST. The strength parallel to the 45SD and BD was the highest in the case when 450 °C aging was done after 750 °C ST.

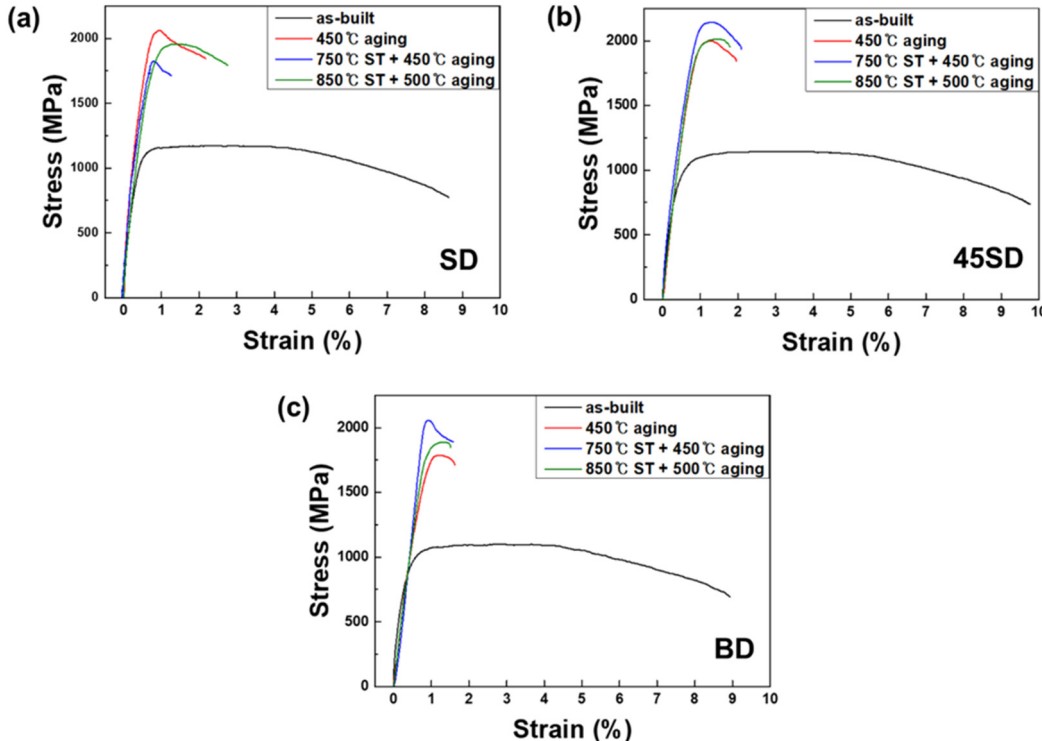

**Figure 10.** Tensile stress–strain curves for maraging steels produced by SLM with different heat treatments, along with direction: (**a**) SD; (**b**) 45SD; and (**c**) BD.

The anisotropic tensile properties of the maraging steels produced by SLM are shown in Figure 11. In the as-built condition, the results demonstrate a clear anisotropy of the yield strength, i.e., the yield strength of the sample was higher in the SD than the other two directions in an aged specimen at 450 °C. The tensile strength of the sample in the as-built condition was almost isotropic. When the samples were aged at 450 °C without ST, both the yield and tensile strength depended significantly on the tensile direction. The SD and 45SD samples showed markedly higher strength than the BD sample in this case. The strong anisotropy of the as-built sample and the sample aged at 450 °C without ST can be easily understood by considering the high degree of crystallographic texture of these samples. However, the sample aged at 450 °C after 750 °C ST also showed a pronounced anisotropy even though the texture was significantly weakened by this heat treatment. In this case, both yield and tensile strengths were significantly lower in the SD than the other two directions. This indicates that the anisotropy of the maraging steel produced by SLM is not solely from the crystallographic texture but also from the other microstructural factors such as the grain morphology. The samples aged at 500 °C after 850 °C ST showed much less anisotropy in comparison with the sample aged at 450 °C after 750 °C ST. On the other hand, the grain morphology was changed significantly by the heat treatment in this case, as shown in Figure 9d. This observation supports the fact that the strong mechanical anisotropy of the maraging steel produced by the SLM process is associated with both the unique grain morphology as well as the strong crystallographic texture. Moreover, it is revealed that the strength of the maraging steel is influenced mainly by the precipitation hardening rather than the cell structure. Although the sample only aged at 450 °C without ST showed much finer cell size than the sample aged at 500 °C after 850 °C ST as shown in Figure 6, the tensile properties after these two different heat treatment conditions showed comparable strengths.

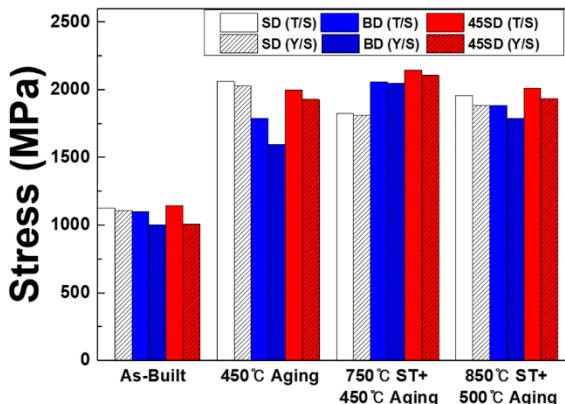

**Figure 11.** Tensile direction dependencies of yield strength (Y/S) and tensile strength (T/S) for maraging steels produced by SLM with different heat treatments.

The results presented in Figures 10 and 11 indicate that a different heat treatment condition has to be considered to obtain the highest strength of the maraging steel fabricated by SLM. If the highest mechanical strength is required in a direction parallel to the SD direction, the maraging steel should be heat treated only by aging at 450 °C without ST. However, 450 °C aging followed by 750 °C ST would be favorable if the final component is expected to bear the maximum applied load in the BD direction. Another important characteristic of the maraging steel produced by SLM is that the highest strength can be achieved by aging and solution heat treatment temperatures lower than commonly used temperatures, i.e., 500 and 850 °C for aging and solution heat treatments, respectively. This behavior is consistent with the hardness changes observed in Figure 3, where the highest hardness could be achieved using heat treatment temperatures significantly lower than the commonly used temperatures. This is presumably due to the very fine cell structure of the maraging steel produced by SLM, as shown in Figure 6a, because the cell boundaries can act as the preferred nucleation sites for precipitates. A finer microstructure in the maraging steel produced by SLM than that produced by conventional methods can provide more nucleation sites for the precipitates and, consequently, can lead to a lower aging temperature for the precipitation hardening.

Fractography was performed after the tensile tests using SEM and no noticeable difference was observed between the samples fractured in different directions. Figure 12 shows the comparison of the fractographic SEM images of the BD samples with different heat treatment conditions. The results are somewhat similar to the previous studies [4,9], that the as-built sample reveals a ductile fracture mechanism with dimple-like features, whereas mixed ductile and quasi-cleavage fracture surfaces are observed in all the heat-treated samples. Moreover, the three differently heat-treated samples have very similar fractographies, as shown in Figure 12b–d. This, together with the observation that the elongation at break does not rely much on the heat treatment conditions, as shown in Figure 10, reveals that the same failure mechanism is in operation in all the three samples regardless of the heat treatment condition. The fracture surfaces of the heat-treated samples show a large number of coarse spherical precipitates, as indicated by white arrows in Figure 12. The spherical precipitates appearing in the fracture surface were larger in Figure 12d than in Figure 12b,c, indicating the coarsening of spherical precipitates during the heat treatment at high temperature. In Figure 12c,d, some precipitates as large as 50 μm are also observed in the fractography, which are apparently much larger than the precipitates observed in the cross-sectional micrographs shown in Figures 4 and 5. One possible reason for this apparent difference is that the cracks are generated along the matrix/precipitate interfaces, which can make the whole precipitate visible on the fracture surface and the apparent size of the precipitates larger than in the cross-sectional images. This is indicative of cracks propagating along the coarse spherical precipitates, which is presumably responsible for the brittle fracture behavior of the heat-treated samples.

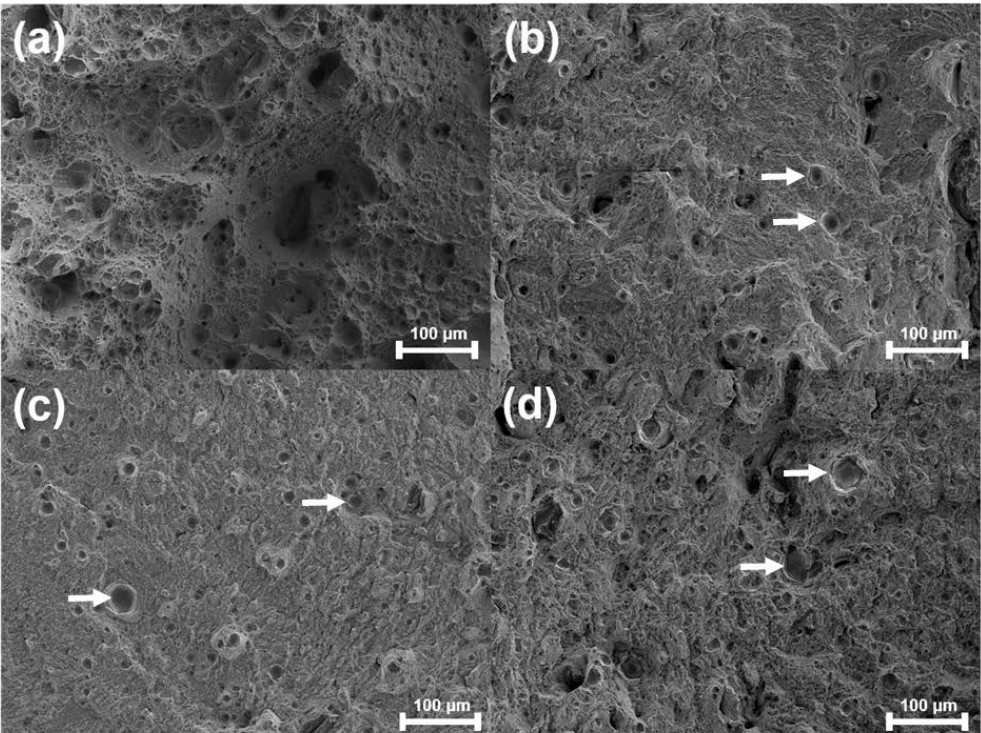

**Figure 12.** SEM fractographic images of BD tensile samples (×200): (**a**) as-built condition; (**b**) after 6 h aging at 450 °C without ST; (**c**) after 750 °C ST for 2 h and aging at 450 °C for 6 h; and (**d**) after 850 °C ST for 2 h and aging at 500 °C for 6 h.

## 4. Conclusions

In this work, experiments were carried out to study the influence of heat treatment conditions on microstructure and mechanical anisotropy of maraging steel produced by selective laser melting. The main findings can be summarized as follows.

The micro Vickers hardness of the maraging steel in the as-built state was around 340 Hv and could be increased by the aging heat treatments, up to around 600 Hv. It was found that the maraging steel can be fully hardened with an aging temperature of 450 °C, which is significantly lower than the conventionally used aging temperature of 550 °C for this alloy. It was also found that the alloy can be hardened by only aging without the solution heat treatment.

The microstructure of the alloy in the as-built state was almost fully martensite due to the rapid cooling. The crystallographic texture in the as-built state showed a strong <100> alignment in the SLM building direction. The texture remained strong when only the aging heat treatment was done without the solution heat treatment. When the solution heat treatment was done before the aging, the texture was significantly weakened.

The tensile test results of the maraging steels heat treated with different conditions indicated that a different heat treatment condition has to be considered to obtain the highest mechanical strength of the maraging steel fabricated by SLM. It was found that the highest strength can be achieved by aging and solution heat treatment temperatures lower than commonly used temperatures, i.e., 500 and 850 °C for aging and solution heat treatments, respectively. In the direction parallel to the laser scanning, the highest ultimate tensile strength was obtained when 450 °C aging was done without solution heat treatment. In the other two directions tested, i.e., directions normal to the building and 45 degrees to the laser scanning direction, the highest tensile strength was obtained when aging was done at 450 °C after 750 °C solution treatment.

Fractography of the maraging steels produced by SLM revealed that, when the steel is fully hardened, the same failure mechanism is in operation regardless of the heat treatment conditions.

Cracks always propagated along the coarse spherical precipitates, which is presumably responsible for the brittle fracture behavior of the alloy in the fully hardened state.

**Author Contributions:** Conceptualization, D.K. and W.L.; Investigation, D.K., J.B.J., and Y.P.; Methodology, T.K., K.H., and W.L.; Resources, J.-J.O.; Supervision, Y.P. and W.L.; Writing—original draft, D.K.; Writing—review and editing, J.-J.O., J.B.J., Y.P., and W.L. All authors have read and agreed to the published version of the manuscript.

**Funding:** This work is supported by the Korea Agency for Infrastructure Technology Advancement (KAIA) grant funded by the Ministry of Land, Infrastructure and Transport, Republic of Korea (Grant 19CTAP-C151899-01).

**Conflicts of Interest:** The authors declare no conflict of interest.

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
