# Peer review of "Effect of Heat Treatment Condition on Microstructural and Mechanical Anisotropies of Selective Laser Melted Maraging 18Ni-300 Steel"

_metals, doi:10.3390/met10030410_

Round 1

Reviewer 1 Report

This paper describes microstructures and tensile properties of selective laser melted maraging steel heat-treated at various conditions. This include systematic results on the effect of heat treatments on mechanical properties, whereby the reviewer believes it would be worth for publication in Metals. But the reviewer would like the authors to consider a couple of minor comments for revising the manuscript.

  1. Could you consider any suggested phases of precipitates using XRD profiles of 450oC aged sample (Fig. 7)? Its profiles exhibit a couple of reflections corresponding to the precipitates. These reflections may suggest any candidates of precipitated phases in terms of crystal structure. Its related comments could be helpful for readers to understand microstructures.
  2. The fracture surface shows a few holes with a size of approximately 50 mm (Fig. 11). The size is somewhat different from the observed size of Ti-rich inclusions (as shown in Fig. 6). The difference seems inconsistent with the fracture mechanism proposed in page 11. More explanation would be needed.

Author Response

The authors thank to the editor and the reviewers for their careful read and thoughtful comments on our paper, which helped very much in improving the manuscript. We have carefully taken the comments into consideration in preparing our new manuscript. In the new manuscript, revised texts are marked with yellow background. The comments are given in the following sections in bold face, while the authors' responses are given in normal font.

Comment of Reviewer #1:

  1. Could you consider any suggested phases of precipitates using XRD profiles of 450oC aged sample (Fig. 7)? Its profiles exhibit a couple of reflections corresponding to the precipitates. These reflections may suggest any candidates of precipitated phases in terms of crystal structure. Its related comments could be helpful for readers to understand microstructures.

Answer: We agree with the reviewer’s opinion but it should be pointed out that there is a practical difficulty on determining the type of precipitates from the XRD. The possible precipitates in the system are known as Ni3Al, Ni3Ti, Ni3Mo and Ni3Fe. These precipitations have main diffraction peaks very close to each other. Therefore, accurate determination about the precipitation type in the XRD is very difficult. In the revised paper, all the possible precipitates are clearly mentioned in the figure 8 (instead of “Precipitates” in the former version) and in the text. The positions of the main diffraction peaks are also indicated now in the text. Further detailed analysis on the type of precipitates can be done by TEM study which is out of the scope of the current study and is already available in the literature. We hope that the current version of the manuscript can meet the reviewer’s expectation.

  1. The fracture surface shows a few holes with a size of approximately 50 mm (Fig. 11). The size is somewhat different from the observed size of Ti-rich inclusions (as shown in Fig. 6). The difference seems inconsistent with the fracture mechanism proposed in page 11. More explanation would be needed.

Answer: In the cross-sectional micrographs shown in Figs. 4-7, the size of the spherical precipitate was varied but no precipitate with the size larger than 50 μm was observed. One possible reason for this difference is that the cracks are generated along with the matrix/precipitate interfaces, which can make the whole precipitate visible on the fracture surface and the apparent size of the precipitates larger than in the cross-sectional images. This also support the fracture mechanisms proposed for the heat treated samples. This point is now clearly stated in the revised version of the paper (on the page 12). We appreciate the reviewer for this nice comment.

Reviewer 2 Report

This paper reports a systematic study of the effect of heat treatment on the microstructural and mechanical anisotropies of SLMed maraging steel. The main novelty lies on the preparation of tensile samples, i.e. 3 different tensile directions. Points to be considered:

  • It is surprising that the first paragraph was not deleted before submission.
  • The characteristic cell structure was not characterized or discussed in relation to the strength of the materials at different conditions.
  • 5 is almost a duplicate of Fig. 4. High resolution SEM observations, e.g. SE imaging after etching or ECC imaging after electro-polishing can provide much better/more information, including cell structure, precipitates and retained/reverted austenite.
  • Due to a very low carbon concentration, the martensite may be considered as BCC.
  • It is not clear how large is the area in Fig. 8 for texture measurement since there is no scale bar. Fig. 8a shows a surprising growth direction along the SD (maybe the BD is not marked correctly for this figure). The texture does not seem to be ‘strong’ (max 3) in all cases.
  • 8a-d were referred to when discussing grain growth/rearrangement/morphology, but these figures are not clear.

Author Response

The authors thank to the editor and the reviewers for their careful read and thoughtful comments on our paper, which helped very much in improving the manuscript. We have carefully taken the comments into consideration in preparing our new manuscript. In the new manuscript, revised texts are marked with yellow background. The comments are given in the following sections in bold face, while the authors' responses are given in normal font.

Comment of Reviewer #2:

  1. It is surprising that the first paragraph was not deleted before submission.

Answer: We are sorry for this mistake. The first paragraph in the introduction part is removed in the revised paper.

  1. The characteristic cell structure was not characterized or discussed in relation to the strength to the strength of the materials at different conditions.

Answer: We have added one figure (Fig.6 in the revised paper) which consists of high magnification SEM images showing the cell structures clearly. However, it was revealed from the SEM images that the cell size is not a dominant factor affecting the strength of the samples. The cell size was the smallest in the as-built sample and became slightly larger with the heat treatments when only aging at 450°C was done without ST. The samples that underwent the ST before the aging heat treatment did not show clear cell structure any more. When comparing these results with the stress-strain behaviors, it is believed that the strength of the samples are mainly influenced by the precipitation hardening not by the cell size.

The discussions corresponding to the microstructure and cell size is now given in the text, on the page 6. Additional discussions regarding the relationship between the anisotropic tensile properties and the microstructure are given in the page 11.

  1. 5 is almost a duplicate of Fig.4. High resolution SEM observations, e.g. SE imaging after etching or ECC imaging after electro-polishing can provide much better/ more information, including cell structure, precipitates and retained/reverted austenite.

Answer: We appreciate the reviewer for this nice recommendation. We have added high magnification SEM images in the revised paper. The paper is now providing discussions regarding the microstructure in more detail with the cell structure.   

  1. Due to a very low carbon concentration, the martensite may be considered as BCC.

Answer: We agree with the reviewer and have replaced the term body-centered tetragonal to “body-centered cubic” in the revised manuscript.

  1. It is not clear how large is the area in Fig. 8 for texture measurement since there is no scale bar. Fig. 8a shows a surprising growth direction along the SD (maybe the BD is not marked correctly for this figure). The texture does not seem to be ‘strong’ (max 3) in all cases.

Answer: To address the reviewer’s concern, we have added the scale bar additionally in the figure. Regarding the Fig.8a, we appreciate very much the reviewer for pointing out our mistake. The former image of the Fig.8a was mistakenly inserted in the wrong direction. We have corrected this mistake and the revised figure has now all the images in the correct direction. The term “strong” was replaced by “clear” or deleted in the revised paper.

  1. 8a-d were referred to when discussing grain growth/rearrangement/morphology, but these figures are not clear.

Answer: We have completely revised the figure to clearly represent the images in the revised version of the paper. The figure now represents the EBSD IPF maps in much larger and clearer than the former version of the figure. We hope that the current version of the figure can meet the reviewer’s expectation.

Round 2

Reviewer 2 Report

The manuscript has been improved. In the discussion of strength contribution, it is clear that precipitates play a major role. However, the contribution from the cell structure should not be overlooked. It is actually the presence of this cell structure that leads to the conclusion that a lower aging temperature (with/without a low solution treatment temperature) may be used. More justification of the result is suggested.

Author Response

The authors thank to the reviewer once again for the second round review of our paper, which helped very much in improving the manuscript. We have carefully taken the comment into consideration in preparing our new manuscript. In the new manuscript, revised texts are marked with yellow background. The comment is given in the following section in bold face, while the authors' response is given in normal font.

Comment of Reviewer :

  1. The manuscript has been improved. In the discussion of strength contribution, it is clear that precipitates play a major role. However, the contribution from the cell structure should not be overlooked. It is actually the presence of this cell structure that leads to the conclusion that a lower aging temperature (with/without a low solution treatment temperature) may be used. More justification of the result is suggested.

Answer: We appreciated the reviewer’s comment and we agree that the fine cell structure is the main reason for the lower aging temperature for the maraging steel produced by the SLM. We have addressed this now clearly in the revised manuscript, on page 11, 321st line.